# Is Neuronal Fatigue the Cause of Migraine?

**DOI:** 10.3390/brainsci12050673

**Published:** 2022-05-20

**Authors:** Michael Pusch, Paola Gavazzo

**Affiliations:** Biophysics Institute, National Research Council, 16149 Genova, Italy; paola.gavazzo@ibf.cnr.it

**Keywords:** migraine, familial hemiplegic migraine, cortical spreading depression, energy expenditure, ion gradients, synaptic activity

## Abstract

The pathological basis of migraine is not fully understood. Familial hemiplegic migraines (FHM) are monogenic forms of severe migraine, caused by mutations in genes encoding various neuronal and/or astrocytic ion transporting proteins. The leading hypothesis regarding the mechanism underlying migraine in FHM is that enhanced electrical excitability leads to increased extracellular potassium levels with subsequent cortical spreading depression. In this short commentary we would like to propose an additional mechanism distinct from enhanced electrical excitability per se. Rather, we propose that FHM mutations cause substantially increased energy expenditure of neurons for re-establishing ion gradients and/or for increased synaptic activity, a mechanism we call neuronal fatigue. Such a metabolic mechanism had been proposed earlier for common migraine and has received recent experimental evidence in particular for the case of FHM3. The hypothesis could be tested in future studies of FHM related models that would need to take metabolic parameters into account.

Migraine is a severe form of headache, which affects a large percentage of the human population with women being more afflicted than men. It is a painful and debilitating neurological condition, which is often pharmacologically difficult to control [1,2]. The pathological mechanisms underlying migraine are likely diverse. Generally, two different hypotheses have been put forward to explain the basis of migraine: the vascular hypothesis, which focuses on the observed dilation of blood vessels as the migraine trigger, and the neuronal hypothesis that focuses on alterations of neuronal excitability as an upstream triggering event [3].

Significant evidence in favor of the neuronal hypothesis was provided by rare monogenic forms of severe migraine, called familial hemiplegic migraine (FHM). Three different forms of FHM have been described. FMH1 is caused by gain-of-function mutations in CACNA1A, which encodes the presynaptic voltage-gated Ca^2+^ channel CaV2.1 [4,5]. FHM2 is caused by loss-of function mutations in the ATP1A2 gene encoding the glial-specific isoform of the α2 subunit of the Na^+^/K^+^ ATPase [6,7]. Finally, gain-of-function mutations in SCN1A, which encodes the voltage-gated Na^+^ channel NaV1.1, lead to FHM3 [8,9]. Thus far, investigations of the various migraine mutations in vitro and in knock-in animals have focused on effects on neuronal excitability [10]. For example, FHM1 mice exhibited increased calcium currents in cerebellar granule cells, enhanced neurotransmission at the neuromuscular junction, and enhanced excitatory transmission in cortical synapses [11,12]. In heterozygous FHM2 mice, the level of the α2 subunit of the Na^+^/K^+^ ATPase was reduced to 50%, and defects in glial-mediated clearance of glutamate were found [13,14]. In FHM3 mice, increased excitability has been observed for inhibitory interneurons [15], in agreement with the predominant relevance of NaV1.1 in these cells. Importantly, all FHM mice showed increased susceptibility to cortical spreading depression [11,12,13,14,15,16], which is believed to accompany the phenomenon of aura in migraines with aura [3]. While these alterations of the excitability of the neural network likely play an important role in migraine induction, with extracellular K^+^ accumulation being a possible trigger [17], no clear common neural mechanism emerged from these studies.

Here, we would like to point to another cellular effect that might be common in all FHM migraine phenotypes. The brain contributes with about 20% to the total body energy consumption and the main energy consuming process is the maintenance of ionic gradient across membranes [18]. Not surprisingly, all processes of the brain have to be highly optimized in their energetic efficiency [19,20]. A significant energetic cost is associated with neural firing, especially if it occurs at high frequency [19]. Unavoidably, each action potential causes a certain increase in extracellular K^+^ and in intracellular Na^+^. The K^+^ and Na^+^ concentrations have to be re-established by the Na^+^/K^+^ pump, partially delegated to glia cells employing K^+^ buffering processes [21]. In fact, the Na^+^/K^+^ pump accounts for about 50% of basal brain glucose oxidation [18]. Similarly, synaptic transmission is a costly process, involving the continuous build-up of complex SNARE protein machineries [20]. The cost is high in order to assure the speed and robustness of synaptic transmission.

Now, in addition to alterations in excitability properties per se, all FHM mutations are predicted to decrease the energetic efficiency of the processes in which they are involved. Gain-of-function mutations of CaV2.1 lead to an increase of vesicular neurotransmitter release, increasing the energy demand of the pre-synapse, and also the post-synapse. Similarly, defects of the glial Na^+^/K^+^ pump lower the capacity of glial mediated K^+^ uptake, increasing consequently the metabolic burden on neurons, which have to handle themselves a larger percentage of K^+^ uptake. Finally, gain-of function SCN1A mutations significantly increase the ion fluxes during a single action potential [22,23], again increasing metabolic demand. In this regard, Hu et al. found that the kinetic properties of K^+^ and Na^+^ channels in fast spiking interneurons are exquisitely tuned to maximize energetic efficiency [24]. It can thus be predicted that in certain stressful situations, for all three FHM genotypes, mitochondrial capacity of affected neurons is not sufficient to sustain the increased energy demand. The exhaustion of mitochondrial capacity could lead to a condition that we call “neuronal fatigue” resulting in complete membrane depolarization and loss of excitability.

Since neuronal fatigue of single neurons increases the chance of fatigue also in neighboring neurons, we hypothesize that, at least in familial hemiplegic migraine, neuronal fatigue might underlie the processes of cortical spreading depolarization and K^+^ accumulation. To test this hypothesis, we propose to focus in future studies of animal FHM models on metabolic aspects, in particular on neuronal energy consumption upon stimulation of elevated electrical activity, see [25].

What about common migraine? A connection between migraine and brain metabolism has been suspected since a long time [26,27]. Mostly clinical evidence suggested that migraine is a response to reduced energy available to the brain [27]. For example, experimentally induced hypoxia in patients suffering from migraine with aura was able to trigger migraine and aura attacks [28,29]. A predictable effect of hypoxia induced metabolic reduction is a decrease of cellular ATP, which can be expected to lead to an activation KATP channels. This might be related to the induction of migraine with and without aura by the KATP opener levcromakalim [30,31].

However, most previous studies have focused on the effects of hypoxia on the antioxidant capacity of the brain, hypothesizing that reactive oxygen species are mostly responsible for migraine generation [27].

In contrast, we speculate that neuronal fatigue, i.e., neuronal depolarization caused by excessive neuronal energy consumption or insufficient energy supply, might be the initial migraine triggering event. Neuronal fatigue of a sufficient number of neurons could then be followed by substantial mitochondrial ROS production, causing migraine associated pain (e.g., via calcitonin gene-related peptide [27]). In parallel, under certain conditions, neuronal fatigue would lead to cortical spreading depression, underlying aura. To test the hypothesis, one would need to measure in sufficient temporal and spatial resolution neuronal membrane potential, neuronal energy consumption, extracellular K^+^ concentration, and pain sensation, clearly not a simple task. Studies of neuronal energy consumption in animal models of familial hemiplegic migraines could be a first step.

## Data Availability

Not applicable.

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
