# Peer review of "Is Neuronal Fatigue the Cause of Migraine?"

_brainsci, 2022, doi:10.3390/brainsci12050673_

Round 1
Reviewer 1 Report
Pusch et al present a paper that proposes an alternative hypothesis for CSD in FHM and probably in "normal migraine". I'm intrigued by this novel hypothesis that could inspire other migraine researches to develop future investigation especially in animal models.
As hypoxia is being discussed in the paper I encourage also to include very recent literature on hypoxia being the trigger for migraine attacks by Frank et al (Migraine and aura triggered by normobaric hypoxia.) Cephalalgia 2020.
Author Response
Thanks, we have incorporated the reference.
Reviewer 2 Report
Dear editor and authors,
It's great to see short papers which introduce new potential mechanisms. I would recommend the authors to briefly look at the papers listed in the link below concerning administration of ATP-sensitive potassium channel opener levcromakalim, and see how this influence your proposed mechanism, neuronal fatigue.
https://pubmed.ncbi.nlm.nih.gov/?term=Al-Karagholi%2C+M.+A-M+and+levcromakalim&size=200
Author Response
Thanks, we have incorporated a sentence related to the effect of KATP openers.
Reviewer 3 Report
Dear Editor,
This is a relatively new perspective of the migraine pathophysiology related to familial hemiplegic migraine types. 'Neuronal fatigue' is an assertive term as the described situation in this opinion manuscript is related to both neurons and astrocytes. In regard to neuronal energy insufficiency, actually, astrocytes have more significant effects. Authors may read this article and may broaden their discussion by including them in the manuscript (Inadequate Brain Glycogen or Sleep Increases Spreading Depression Susceptibility. Ann Neurol. 2017 Dec 15. doi: 10.1002/ana.25122.).
Mitochondrial insufficiency may also be important in cortical spreading depression as mentioned in a review (Migraine and neuroinflammation: the inflammasome perspective. J Headache Pain. 2021 Jun 10;22(1):55. doi: 10.1186/s10194-021-01271-1.).
From global point of view both energy insufficieny and resulted mitochondrial liabillity may result in neuronal inflammasome activation which eventually causes trigeminovascular activation and headache. This point also published in an article last year in FHM1 mutant mice (Widespread brain parenchymal HMGB1 and NF-κB neuroinflammatory responses upon cortical spreading depolarization in familial hemiplegic migraine type 1 mice. Neurobiol Dis. 2021 Jun 9:105424. doi: 10.1016/j.nbd.2021.105424.).
It would be better for authors to broaden their discussion in line with these literature.
Author Response
We thank the referee for the suggestion. However, it appears that inflammation is a “down stream” event triggered by the primary causes (hypothesized as being caused by neural fatigue). Thus, we rather prefer not to include these in order to maintain the commentary in focus.